# Learning Assessment from a Lecture about Fundamentals on Basic Life Support among Undergraduate Students of Health Sciences

**DOI:** 10.3390/healthcare8040379

**Published:** 2020-10-01

**Authors:** Luis Fernando Barbosa Tavares, Rodrigo Daminello Raimundo, Claudio Leone, Cyntia Souza Carvalho Castanha, Adriana Gonçalves de Oliveira, Blanca Elena Guerrero Daboin, Joseane Elza Tonussi Mendes, Luiz Carlos de Abreu

**Affiliations:** 1Laboratory of Study Design and Scientific Writing, Centro Universitário Saúde ABC, São Paulo 09060-650, Brazil; luisfbtped@gmail.com (L.F.B.T.); rodrigo.raimundo@fmabc.br (R.D.R.); claudio.leone@fmabc.br (C.L.); cyntiacastanha@gmail.com (C.S.C.C.); agdeoliveira@gmail.com (A.G.d.O.); blanca.daboin@fmabc.br (B.E.G.D.); tonussimestrado2016@gmail.com (J.E.T.M.); 2Graduate Program in Medical Sciences, Universidade de São Paulo, São Paulo 05508-060, Brazil

**Keywords:** cardiopulmonary resuscitation, heart arrest, students, health occupations

## Abstract

*Introduction*: Cardiac arrest is one of the leading public health problems worldwide and in Brazil. A victim of cardiorespiratory arrest needs prompt basic life support (BLS) to increase survival. *Objective*: To evaluate the performance of a synthesis lecture on BLS given to university students in Health Sciences. *Methods*: A total of 422 undergraduate students in Nursing, Physiotherapy, and Medicine participated in this study. Data were collected by applying a pre-test through a BLS questionnaire based on the American Heart Association guidelines. *Results*: Students obtained a minimum grade of 40% of the pre-test questions. The score increased to 75% in the post-test; the students with the best performance in the pre-test maintained a higher total number of correct answers in the post-test. There was also better performance in those with previous training in BLS. The students from the first year of medical school were the ones who benefited the most from the lecture. *Conclusion*: Regardless of the grade course, the Health Science students showed a significant improvement in their level of knowledge after attending the synthesis lecture, indicating its adequacy to promote initial learning about BLS.

## 1. Introduction

Globally, cardiovascular disease is one of the leading causes of death [1,2]. In developed countries, such as the United States, more than 500,000 deaths among children and adults occur each year due to cardiac arrest, leading it to be considered one of the significant public health problems [2,3,4,5]. In Brazil, it has estimated that about 200,000 cardiac arrests occur annually [6].

In the event of a sudden illness, an individual who suffers cardiopulmonary arrest (CPA) readily needs the proper application of high-quality basic life support (BLS) to increase survival. There is an association between the number of people trained in BLS and victims of CPA who arrive at the emergency medical service receiving cardiopulmonary resuscitation (CPR). It is estimated that for every minute of CPR without adequate BLS, the victim’s chances of survival decrease by 10% [1,6]. Even within the ideal condition of a CPA, which would be in the hospital and witnessed, the mortality rate is still very high, reaching 80% to 85% [7].

The BLS is structured by a systematic action sequence to be followed in a CPA event. Healthcare students have insufficient knowledge to perform high-quality BLS. There are advances in the field of BLS knowledge and training and specific legislation, including providing means for a greater availability of automatic external defibrillator (AED) in public places [8].

The dissemination of knowledge and the implementation of measures that facilitate appropriate care routines for CPA victims must accompany the progress of science, and the use of simulation in universities of Health Sciences is one of these measures. However, this is something that has been implanted recently in Brazil, and the investment for its realization involves infrastructure, such as technology and trained teachers, which still makes the cost of implementation high [9].

The direct instruction model proves to be suitable for teaching BLS, where it is assumed that students can benefit from well-designed and unambiguous instructions. According to a study carried out in Australia, simulation training improves the professionals’ confidence; however, this does not mean improving the ability to perform CPR procedures [10,11,12,13,14].

The low level of knowledge of Nursing, Physiotherapy, and Medicine students in BLS is proven, which implies not performing high-quality CPR, resulting in a negative social impact. Considering the new teaching proposals on the BLS, this study aims to evaluate the effect of a synthesis lecture on BLS as a teaching strategy among undergraduate students of medical sciences.

## 2. Materials and Methods

### 2.1. Study Design and Location

This is a non-randomized experimental controlled study carried out at three different institutions as follows: Centro Universitário Saúde ABC, in Santo André city and at Faculdades Metropolitanas Unidas, in the city of São Paulo both of them in the State of São Paulo, and at Faculdades Integradas de Patos, in the city of Patos, State of Paraiba, Brazil.

### 2.2. Criteria Eligibility

Undergraduate students from the first and fourth years from Nursing, Physiotherapy, and Medicine courses were included. Those students who did not complete the research protocol were excluded.

### 2.3. Study Sample

It consisted of 422 university students from the first and fourth year of Health Sciences course, 175 Medicine students, 136 from Nursing, and 111 Physiotherapy students. Of them, 281 were in their first year and 141 were in the fourth year of their respective courses (Table 1).

The first and fourth-year students’ samples were independent, as the data were collected simultaneously and not after a follow-up period. First and fourth-year students were chosen to assess students’ knowledge at the beginning and four years later. The study was not randomized; all available students were invited to participate, respecting the sample calculation.

Overall, 78.0% of the students were female and 22.0% were male, which shows a non-normal distribution. The predominant age group was between 18 and 24 years old, which concentrated almost 1/3 of the students. The median age was 21 years.

### 2.4. Experimental Protocol

The method used represents the simulation of two clinical situations of cardiorespiratory arrest. Through an expository class with a practical demonstration, the BLS sequence is reproduced in a synthesized manner. It is done in a short period, grouping all the necessary information to transmit this knowledge’s acquisition and retention in the different phases of the courses. Figure 1 illustrates the steps to assess the learning resulting from a synthesis lecture about BLS in a sample of undergraduate students of health sciences.

A questionnaire on BLS based on the American Heart Association (AHA) was performed both in the pre-test assessment and post-test stages to collect the data. It consisted of 20 multiple choice questions, only one correct alternative, and open questions regarding demographic data. The students were asked if they had previously undergone any BLS training, and in case of an affirmative answer, it informed the data. The same questionnaire was applied in the pre-test and post-test [8].

The data collection instrument consisted of a block of 4 sheets, the first 2 being the copies of the informed consent form (ICF), the next sheet containing the 20 questions on BLS, and the last one corresponding to the feedback questions. The students received the instruction to keep the first copy of the ICF and return the second one, which was signed along with the rest of the block after completing the pre-test. The tests were carried out in the classroom, with the chairs organized in rows to solve the questions individually, and the time spent to perform them was 30 min.

The BLS synthesis lecture lasted 50 min, using as support material two mannequins, a baby, and a child to demonstrate the CPR maneuvers (compressions and ventilations) to students. In addition, an interface (face shield scarf) was given to the students to ventilate the mannequins. During the lecturer, scenarios based on clinical cases were created to simulate the correct CPR sequence, including a demonstration using an automatic external defibrillator (AED). The mannequins were from the Little Family Manikin Pack model (Laerdal Medical Corporation, New York, NY, USA) and the AED (Laerdal Medical Corporation, New York, NY, USA).

The lecture on basic life support was administrated as follows. The Basic life support synthesis class (BLSSC) was carried out within a logical sequence, and initially, the epidemiology of cardiac arrest in Brazil and worldwide was addressed, in addition to the survival rate after CPR, and then guidance on the survival chain was provided.

Next, instructions were given on how to recognize a victim in cardiopulmonary arrest, information about the constituents of the BLS, demonstration of the locations for checking the central pulses according to age groups, a practical demonstration with the mannequin of the correct opening of the airways in traumatized and non-traumatized victims, and how to provide adequate ventilation according to different age groups. Explanations about the locations and mode of cardiac compressions were also given for babies, children, and adolescents with 1 and 2 rescuers present and with and without guaranteed advanced airway.

They were instructed on the characteristics of good quality CPR, where the use of the AED was reproduced and the peculiarities were explained according to the age groups, which were followed by a demonstration of the complete sequence of the CPR using the AED, the duration of the CPR cycles, and the reassessment of the CPR patient.

Afterwards, a discussion of clinical cases was initiated, with the demonstration of the correct sequence of care in two different scenarios, one after finding unconscious child and baby victims and another scenario where a sudden collapse was seen in an adolescent. The BLSSC ends with clarifying students’ doubts before the post-test. Right after the class, the post-test was applied with the same conditions as the pre-test.

Throughout the classroom, the researcher was present to apply the pre-test and post-test questionnaire and provide the synthesis class’s administration. The researcher corrected the questionnaires through the students’ answer template, and the data were stored in a table prepared by the researcher in the Excel^®^ program.

### 2.5. Sample Calculation

The number estimated as optimal was 36 students in each group based on the average observed in a previous study of 35.8% of correct answers [8] with a standard deviation (sd) of 15.0 percentage points and an alpha = 0.05 and a test power of 0.80. It aims to discriminate between correct answers in the pre-test and the post-test of at least 10 percentage points.

Assuming a possible loss of 10% in each group, between the first and the second assessment, we added four individuals to the sample size estimated initially, which resulted in 40 students per group in the first test, in order to guarantee at least 36 students in each group for the final analysis. The total estimated minimum required was 216 students. The estimate was performed using the Medcalc 17.2 software (Medcalc, Ostend, Belgian).

### 2.6. Variables

The variables analyzed were the total number of correct answers in the pre-test and post-test, the difference in the number of correct answers between the pre-test and post-test (dependent variables), the group of student’s course, their grade course, and the presence of previous CPR training (independent variables).

### 2.7. Sample Loss

There was a sample loss of 28 students (6.6%) due to not complying with the protocol as planned.

### 2.8. Statistical Analysis

The data were described considering distributions by relative and absolute frequency and measures of central tendency: average, median, and fashion. The analytical treatment was performed by comparing means or medians using the Mann–Whitney test (independent data), Wilcoxon test, and Kruskal–Wallis test (non-independent data). Besides, correlation analyses were performed by Pearson’s coefficient.

The level of significance was adopted (α) as 0.05. The interquartile range compared the medians of the pre-tests and post-tests between first and fourth-year students of Medicine, Nursing, and Physiotherapy. The difference in correct answers between the pre-test and post-test had a normal distribution with a median of 6.5 points. For data analysis, the following software was used: MedCalc version 17.2 and GraphPad version 6.01 (GraphPad Software, San Diego, CA, USA).

### 2.9. Ethical Aspects

The Ethics Committee of the Faculdade de Medicina do ABC approved the study with the number 557–716 on 12 March 2014. All research participants signed an informed consent form.

## 3. Results

Of the 422 students included in the final sample, 186 (44.1%) had previously undergone BLS training. In the knowledge assessment, 422 students obtained a median of correct answers in the pre-test of 40.0%. In contrast, in the post-test, the median was 75.0%, *p* < 0.001, as shown in Figure 2.

The students who had received prior training in BLS improved their performance between the pre-test and post-test (Figure 3) significantly.

Upon analyzing the groups of students, regardless of whether they had undergone prior training, there was no difference in the evolution of the post’s scores and the pre-test for the two groups (Figure 4).

The total number of correct answers between pre-test and post-test in all groups showed an increasing trend. This fact is corroborated by students’ performance with the highest number of correct answers in the pre-test and evolved with an improvement in the post-test score. There was no statistically significant difference in comparing the medians of the correct answers in the respective courses (*p* = 0.1117).

On comparing the medians of correct answers in the post-test of the students of the respective courses, there was a statistical significance (*p* < 0.001) between Medicine and Nursing (*p* < 0.001), as well as between Medicine and Physiotherapy (*p* < 0.001). However, there was no statistical difference when comparing the Physiotherapy and Nursing courses (*p* > 0.05). Table 2 illustrates the correct answers median in the pre-tests and post-tests and the median differences of correct answers.

Table 3 illustrates a statistically significant difference between the first and fourth-year students in the pre-test of the three courses: Nursing, Physiotherapy, and Medicine; only the nursing course did not show a difference between the first and fourth-year students in the post-test. The only course whose students showed a statistically significant difference in the comparison of performance between pre-tests and post-tests was Medicine.

According to their number of correct answers in the post-test, of the 422 students evaluated, 411 (97.5%) improved performance. 2.7% of students worsened or continued with the same previous knowledge

## 4. Discussion

It was observed that the synthesis class model on the fundamentals of BLS offered to the 422 students of health sciences was satisfactory. These findings indicated cognitive learning, since most undergraduate students in Nursing, Physiotherapy, and Medicine practically doubled their indicators comparing pre-test and post-test evaluations. The median of their post-test results approached the performance result recommended by the American Heart Association (AHA) standard course on BLS (84%).

In Pakistan, a study showed that knowledge of first aid is something that medical students have not given due importance with a consequent effect on the quality of performing CPR for newly qualified doctors; despite the importance of training, BLS now has worldwide recognition [15].

The prior knowledge of Health Sciences students about BLS was 40%, which is similar to the findings of a study on university students’ knowledge about BLS. Correct answers were frequent for 35.8% of questions about BLS [8].

In France, a study carried out with medical students from more advanced years showed that they do not feel prepared to conduct BLS and that only 1/3 of the students reported being able to provide this assistance. Using an instrument with objective self-perception questions about BLS, more than 2/3 of the students reported not being qualified or having insufficient qualification, while only 2% reported feeling qualified [7].

In this research, trained students and those without previous BLS training obtained low pre-test scores that were below the 84% recommended AHA score. These findings point to a cognitive deficit on the necessary procedures for adequate high-quality BLS, denoting the need for permanent and continuing education on this significant theme.

Studies comparing the theoretical performance between students with and without prior training showed better theoretical results than previous training. These results converge to those of the present research, indicating that previous training indicates students’ better BLS [15,16,17,18].

In this study, students from the first year of Medicine showed a significant difference from the initial and final evaluations. They were the group that obtained the highest benefit of the BLSSC. The first-year nursing students were the group with the most significant number of individuals who reported having previous knowledge about BLS. These students retained satisfactory performance in the post-test but with a smaller difference in correct answers between the pre-test and post-test; that is, the group benefited less from the class.

In India, a study evaluated the retention of knowledge and skills transmitted to first-year medical students about BLS. The pre-test showed unsatisfactory results but improvement in the post-test after demonstration training. It corroborates with this current study’s results, which showed improved cognitive knowledge in the three courses after the BLS lecture [19]. Other studies carried out with first-year students showed results similar to those of the present study [15,17,20].

The analysis of students’ pre-test performance from the fourth year of the Nursing, Physiotherapy, and Medicine courses showed that the theoretical knowledge on BLS was low, indicating insufficient knowledge, even in these university students of a more advanced course level.

Overall, students from the fourth year of the courses performed better in the post-test, which was attributed to previous training outside the university environment or BLS activities. Correctly, as for the nursing course students, who obtained excellent performance both in the pre-test and in the post-test, these results can be attributed to the conditions said before and the fact that most of them were technical professionals in the field of Nursing.

Our study reveals that the first-year students had the highest gain concerning previous knowledge. However, they were not the ones who had the highest score in the final evaluation, which is attributed to more advanced students with previous training, emphasizing the retention of knowledge in the face of exposure.

A study reported that young doctors do not perform adequate CPR due to university teaching failure. Their opportunities for training in the workplace were remote, showing that BLS education strategies should be reviewed at universities [21]. In another study in Romania, they share the idea of including BLS training in the university curriculum [22].

According to this study’s protocol, the demonstration of a BLS lecture aims to provide prior instruction on the execution of motor skills, enabling the student to obtain information about the nature of the task to be performed, evidencing the “how to do” process. The purpose of these lectures is to stimulate students at the beginning of the course in the cognitive part and as for students of a more advanced level to contribute to the reinforcement of knowledge, since cognitive learning is a premise for the future development of psychomotor skills.

According to AHA guidelines on medical education, the literature on resuscitation education has been limited to results that focus on a short learning period [2]. With the model of synthesis BLS lecture presented in the current study, it was possible to achieve positive results compared to those in the literature, proving to be very relevant, especially for students at the beginning of the course.

A study carried out in the United Kingdom reported that students improved on the post-test; these findings are similar to the present study results. However, on analyzing the students who worsened or continued with the same previous knowledge, there was a smaller number (2.7%) in our study than the UK research, where 29% worsened [18].

University students in the health field do not have enough knowledge about BLS to perform high-quality CPR. It is a profound reflection of the failure to teach this topic in universities. These institutions need to invest in better realistic training centers and continuing education, making it possible to improve knowledge retention and transmit confidence to the student, who currently takes a deficit of knowledge from the university to practical life.

In the sample of this study, the performance of students of health sciences was sufficient to show learning in the cognitive part from the BLS lecture’s execution. Most students showed improvement in their indicators when comparing between the pre-test and post-test assessments. The students who benefited the most from the training were first-year Medical students. However, the highest grades were obtained by the fourth-year Medical students. The undergraduate course that benefited the least was the Nursing course, where the first-year students demonstrated more excellent prior knowledge about BLS.

BLS’s traditional teaching method based on translational science is the best to convey confidence to the student. The complete course is based on realistic simulation, allowing the student to acquire cognitive skills and develop practical skills. However, the synthesis lectures on BLS might be able to find its education space at different times for each course in order to improve cognitive learning functions with more frequent training, having favored the short time of accomplishment.

To the best of the author’s knowledge, no studies compare the performance of this type of educational activity in undergraduate students of Health Sciences in three different institutions in Brazil’s two distant regions. The north region and southeast region have significant differences in human development. Furthermore, this initiative is economically viable with an excellent cost–benefit ratio for the universities, considering that the traditional method training on BLS following AHA guidelines requires eight hours. The contribution of new information and communication technologies in the teaching-learning process for training in health, as well as the reflection on distance education and its concepts, differentiating it from the concepts of remote methodology and the use of technologies [23].

Our findings showed a significant improvement in cognitive learning and provided initial learning about BLS for undergraduate students.

## 5. Conclusions

First-year students benefited the most from the lecture; their performance was proportionally better in all the evaluated courses. The medical course students proportionally showed the highest increase in the level of knowledge on BLS support.

Overall, regardless of the grade course, most students improved their knowledge after the proposed synthesis lecture. The tendency of improvement was obtained, indicating the synthesis lecture’s adequacy to promote initial learning on basic life support for undergraduate students in the area of Health Sciences.

## Figures and Tables

**Figure 1 healthcare-08-00379-f001:**
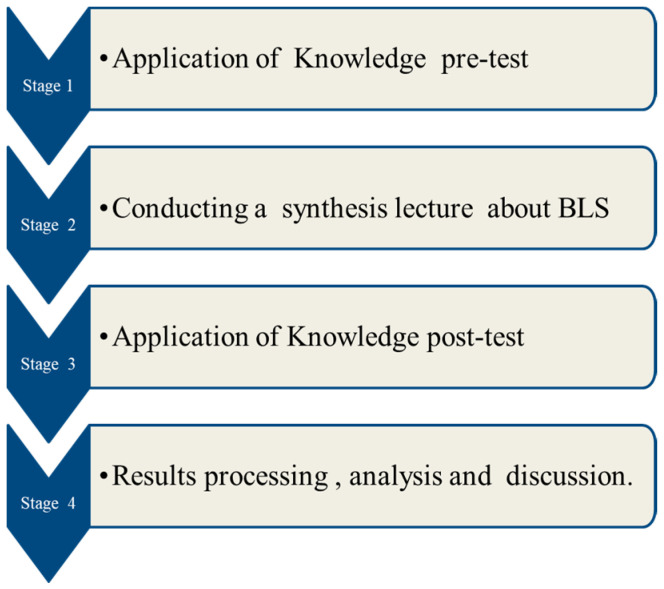
Stages of the research protocol.

**Figure 2 healthcare-08-00379-f002:**
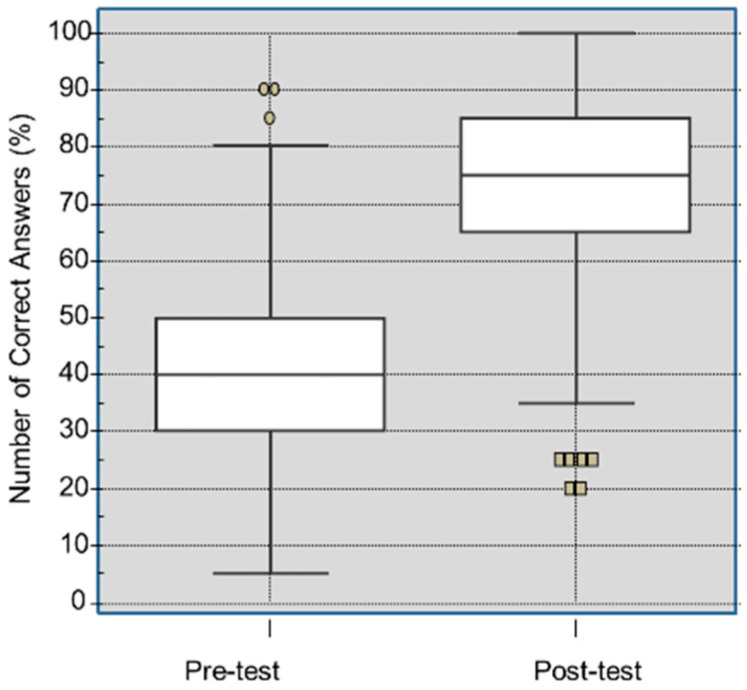
Distribution of students according to the percentage of correct answers in the pre-test and post-test (Wilcoxon test, *p* < 0.001). The difference in correct answers between the pre-test and the post-test had a normal distribution, with a median of 6.5 points. Points: pre-test outliers and squares: post-test outliers

**Figure 3 healthcare-08-00379-f003:**
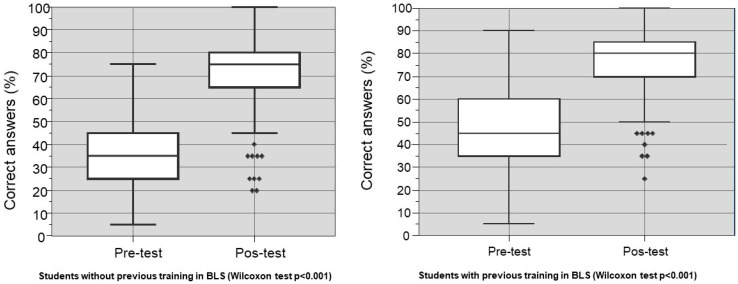
Distribution of correct answers (%) between students with and without previous training in basic life support (BLS) (Willcoxon test *p* < 0.001). Points: post-test outliers and squares: post-test outliers

**Figure 4 healthcare-08-00379-f004:**
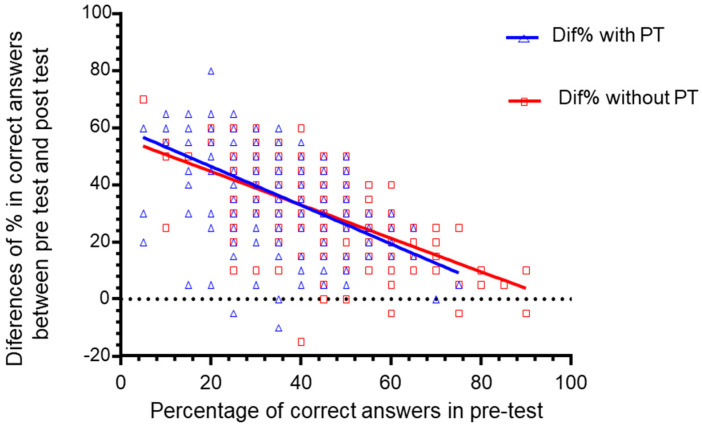
Evolution trend of the difference in the proportion of correct answers between pre-test and post-test. Pearson coefficient r = −0.5203 (IC 95% = −0.6076 to −0.4205) for those with previous training (PT) and r = −0.6043 to 0.5043) for those without PT, both statistically significant (*p* < 0.001), Comparison of correlation (Coefficient of the groups with and without training (Z = 1.2467). The dotted line of the zero value discriminates against those who performed positively, those who did not improve and/or worsen knowledge

**Table 1 healthcare-08-00379-t001:** Distribution of the research population by course and grade attended.

	Course	Medicine *n* (%)	Nursing *n* (%)	Physiotherapy *n* (%)	Total *n* (%)
Grade					
First Year		132 (75.4)	89 (65.4)	60 (54)	281 (66.6)
Fourth Year		43 (24.6)	47 (34.6)	51 (46)	141 (33.4)
Total		175 (100.0)	136 (100.0)	111 (100.0)	400 (100.0)

**Table 2 healthcare-08-00379-t002:** Comparison of the distribution of correct answers between students from the first and fourth year, in the pre-test and post-test and the difference in performance between both tests (Mann–Whitney test).

Grade	Median Pre-Test	CI 95%	*p*-Value	Median Post-Test	CI 95%	*p*-Value	Median of the Difference of Correct Answers Post-Pre-Test	*p*-Value
First year	7 (35%)	(7–7) (35–35%)	*p* < 0.001	15 (75%)	(14–15) (70–75%)	*p* < 0.001	7 (35%)	*p* < 0.001
Fourth year	10 (50%)	(9–11) (45–55%)		16 (80%)	(16–17) (80–85%)		6 (30%)	

**Table 3 healthcare-08-00379-t003:** Percentage of correct answers in the pre-tests and post-tests and difference in right answers between them by course and year.

Course	Median of Correct Answers Pre-Test (%)	Interquartile Range (p25 to p75)	*p*-Value Difference between 1st and 4th Year	Median of Correct Answers Post-Test (%)	Interquartile Range (p25 to p75)	*p*-Value Difference between 1st and 4th Year	Median of the Difference of Correct Answers between Post and Pre-Test (%)	Interquartile Range (p25 to p75)	*p*-Value, the Difference between 1st and 4th Year
Nursing 1st year	40	30–50	*p* = 0.0462	70	60–80	*p* = 0.7042 (ns)	25	15–40	*p* = 0.1460 (ns)
Nursing 4th year	50	35–60		70	60–80		25	10–35	
Physiotherapy 1st year	30	22.5–40	*p* < 0.001	70	60–75	*p* < 0.001	32.5	25–47.5	*p* = 0.4450 (ns)
Physiotherapy 4th year	45	40–55		80	80–85		35	20–40	
Medicine 1st year	35	25–45	*p* < 0.001	75	70–85	*p* = 0.007	42.5	30–50	*p* < 0.001
Medicine 4th year	55	41.3–65		85	80–90		30	20–38.8	

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
