# Peer review of "Learning Assessment from a Lecture about Fundamentals on Basic Life Support among Undergraduate Students of Health Sciences"

_healthcare, 2020, doi:10.3390/healthcare8040379_

Round 1

Reviewer 1 Report

This study evaluated the effectiveness of BLS courses for health science students in universities. The ratio of test correctness totally improved from 40% to 75%. Furthermore, in all groups of students, the tendency of the improvement was obtained. This study is interesting, but almost the same as previous studies. Thus, they need to represent the de-novo views compared with them.

 [Methods]

Please clearly show the primary endpoint in this study.

Line 90: Why did the number of fourth-year students be smaller than one of first-year?

Line 101: There is the typo of “basil”.

Line 125: In what country is the AED produced?

Line 147: The discussion with students in BLS course can give a bias of the post-test. Please refer to this in Discussion section.

Line 165: In what country is the software produced?

Line 186: The authors described the result of the difference. Please move it to Result section.

[Results]

Line 199: The authors need not to describe “87.5%”, because it is enough to show 40.0% and 75.0%.

Line 207 (Fig. 3): Is there the significant difference of correct answer (%) of pre-test between students with and without BLS training?

Line 227 (table 2): The authors need to remake the table. They should show the pretest results using % such as other results.

Line 236 (table 3): They should rewrite p=ns instead of p=0.0462 and others.

[Discussion]

Line 330: should change PCR to CPR.

Line 344: what is ASSBV?  

I would like to know the duration between this trial and previous experienced BLS training.

Author Response

Dear Dr.,

Thank you very much for your relevant revision of our submission. We have revised the material to eliminate the issues raised. The added or modified words, phrases, and sentences are in red. We hope the present version can be accepted.

  • English language and style

(x) Extensive editing of English language and style required
( ) Moderate English changes required 
( ) English language and style are fine/minor spell check required 
( ) I do not feel qualified to judge about the English language and style

Our answers to your questions, in red as follows:

  • This study evaluated the effectiveness of BLS courses for health science students in universities. The ratio of test correctness improved from 40% to 75%. Furthermore, in all groups of students, the tendency of the improvement was obtained. This study is interesting, but almost the same as previous studies. Thus, they need to represent the de-novo views compared with them.

 Answer:

The study's differential is the method used; this teaching model adapted to university students can transmit knowledge without previous training in basic life support and reinforce it when they already had it. Furthermore, this alternative is economically viable, with an excellent cost-benefit ratio for the universities.

 [Methods]

  • Please clearly show the primary endpoint in this study.

Answer

The method used represents the simulation of care in 2 clinical situations of cardiorespiratory arrest, in which, through an expository class with practical demonstration, the sequence of basic life support is reproduced in a synthesized way, in a short period, grouping all the necessary information to transmit the acquisition and retention of this knowledge in the different phases of the courses.

  • Line 90: Why did the number of fourth-year students be smaller than one of first-year?

Answer

The study was not randomized; all available students of the fourth years were invited to participate, respecting the sample calculation.

  • Line 101: There is the typo of “basil”.

about basil life support (SLBLS) in the sample of undergraduate students of health sciences.

Answer

about basic life support (SLBLS) in the sample of undergraduate students of health sciences.

  • Line 125: In what country is the AED produced?

Answer

Produced by Laerdal Medical corp. with headquarters in the USA, Canada, and Brazil

  • Line 147: The discussion with students in the BLS course can give a bias of the post-test. Please refer to this in the Discussion section.

Answer

During the lecture, the sequence of basic life support (BLS) was reproduced and demonstrated with the use of the mannequins and the automatic external defibrillator. After the class, the pertinent doubts concerning the clinical cases were solved,  since most of the students, mainly those of the first year had never had contact with the subject.

  • Line 165: In what country is the software produced ?

Answer

Belgium

  • Line 186: The authors described the result of the difference. Please move it to the Result section.

The difference in correct answers between the pre and the post test had a normal distribution, with a median of 6.5 points.

Answer

As requested, this was changed to the results section

[Resultados]

  • Line 199: The authors need not to describe "87.5%" because it is enough to show 40.0% and 75.0%.

In the knowledge assessment, the 422 students obtained a median of correct answers in the pre-test of 40.0%. In contrast, in the post-test, the median was 75.0%, p <0.0001, which represents a median increase of 87.5% in the number of correct answers (figure 2).

Answer

In the knowledge assessment, the 422 students obtained a median of correct answers in the pre-test of 40.0%, while in the post-test, the median was 75.0%, p <0.0001.

  • Line 207 (Fig. 3): Is there the significant difference of correct answer (%) of pre-test between students with and without BLS training?

Answer

We did not initially have this answer because it was not one of the objectives of the study, however, as the reviewer wanted to know, we did the statistical analysis. We concluded that the difference between the pre-test of students with and without prior training was statistically significant.

Student's t-test was performed because both groups had a normal distribution of results and, therefore, it was possible to compare by means and respective standard deviations;

Normality was tested by the D'Agostino-Pearson test whose result was p = 0.5239 (in this test, a p> 0.05 indicates a normal distribution of the 2 groups);

The conclusion is that the averages are statistically different and that the pre-test performance is on average, better (more correct) among the students who had previously done a training (regardless of where and in what form or type of training).

  • Line 227 (table 2): The authors need to remake the table. They should show the pre-test results using % such as other results.

Answer

Modified

  • Line 236 (table 3): They should rewrite p=ns instead of p=0.0462 and others.

Answer

Modified

[Discussion]

  • Line 330: should change PCR to CPR.

Answer

acquisition of cognitive knowledge and excites the first year student by demonstrating a CPR

Line 344: what is ASSBV?  

  • accomplishment, and at a low cost, ASSBV finds its space in education.

Answer

ASSBV is the Portuguese acronym for a synthesis lecture on basic life support, but in English, it corresponds to the SLBLS. It was a typo.

  • I would like to know the duration between this trial and previous experienced BLS training.

Answer

The time elapsed between the previous training of students who reported having it, and most students did not answer the study despite being in the data collection questionnaire.

Reviewer 2 Report

This is a valuable and well-designed study with a potential relevance for medical education. I have only one comment: In the Conclusions the Authors wrote:

"The performance among students of the first year was proportionally higher than that of 350 students from the fourth year, from all the evaluated courses."

In fact, first year students were those who benefited most from the class - this should be modified to improve the clarity of the Conclusions.

Last, English style needs improvement throughout the manuscript.

Author Response

Dear Dr.,

Thank you very much for your relevant revision of our submission. We have revised the material to eliminate the issues raised. The added or modified words, phrases, and sentences are in red. We hope the present version can be accepted.

  • "The performance among students of the first year was proportionally higher than that of 350 students from the fourth year, from all the evaluated courses."
  • First-year students benefited most from the class - this should be modified to improve the clarity of the Conclusions.

Answer

It was observed that the first-year students of the respective courses' performance were considerably superior concerning the students of the fourth year.

Included in the conclusions

Round 2

Reviewer 1 Report

I would say you need to rewrite with the comments below.

The authors need English correction by some native English specialist.   

Line 20-24: The authors need to evaluate grammatical errors.

In Introduction: The number of paragraphs is too much. Please decrease the paragraphs at least half of them.

In line 77: They do not need the abbreviation FMU.

In line 85: It would be better, “Medicine of 175, Nursing of 136 and Physiotherapy of 111”

In Table 1: not “,”, but “.”

In line 105: The word, AHA, has been shown in Introduction.

In line 124: What is country and city about Laerdal Medical?

In line 125-146: They should describe not itemize style but usual style.

In line163/169: They deleted the title of Variables and Sample loss.

In Statistical analysis: As I mentioned, they do not need to make many new lines.

In Results: Also, they do not need to make many new lines.

In Table 2: I am unsure of the meaning of “a” in (7 a 7) and so on. Please rewrite to usual one.

In Table 3: Also, I am unsure of the meaning of “a” in “30 a 50” and so on. Please rewrite them.

In Discussion: Please decrease many paragraphs.

Author Response

Dear Dr.,

Thank you very much for your relevant revision of our submission. We reviewed the material to eliminate the issues raised. The added or modified words, phrases, and sentences are in red. We hope the present version can be accepted.

 [Abstract]

  • In line 20-24: The authors need to evaluate grammatical errors.

Results: Students obtained a median of correct answers for 40% of the questions in the pre-test. In the post-test, the median increased to 75%. The students with the best performance in the pre-test maintained a higher total number of correct answers in the post-test, those with previous training were those who had the best performance, and those in the first year of the medical course were the ones who benefited the most from the class.

Answer

Students obtained a minimum grade of 40% of the pre-test questions. The score increased to 75% in the post-test, the students with the best performance in the pre-test maintained a higher total number of correct answers in the post-test. There was also a better performance in those with previous training in BLS. The students from the first year of medical school were the ones who benefited the most from the lecture.

 [Introduction]

  • In Introduction: The number of paragraphs is too much. Please decrease the paragraphs at least half of them.
  • Line 30-76: The number of paragraphs is too much. Please decrease the paragraphs at least half of them.

Answer

The introduction was reduced. The  paragraphs in red were deleted

Globally, cardiovascular disease is one of the main cause of death [1, 2]. In developed countries, such as the United States, more than 500,000 deaths among children and adults occur each year due to cardiac arrest, leading it to be considered one of the major public health problems [2-5]. In Brazil, it is estimated that about 200,000 cardiac arrests occur annually [6].

In the event of a sudden illness, an individual who suffers cardiopulmonary arrest (CPA), readily needs the proper application of high quality basic life support (BLS), to increase the chance of survival. There is an association between the number of people trained in BLS and victims of CPA who arrive at the emergency medical service receiving cardiopulmonary resuscitation (CPR). It is estimated that for every minute of CPR without adequate BLS, the victim's chances of survival decrease by 10% [1, 6]. Even within the most ideal condition of a CPA, which would be in the hospital and witnessed, the mortality rate is still very high, reaching 80 to 85% of the cases [7].

The BLS is estructured by a sequence of systematic actions to be followed in the ocurrence of a CPA event. Healthcare students have insufficient knowledge to perform high-quality BLS. There are advances in the field of BLS knowledge and training, as well as in specific legislation, including providing means for better availability of automatic external defibrillator (AED) in public places [8].

It is noteworthy, then, that BLS should be taught to all students of the health field, as this is an expectation of patients and the lay public, regardless of the level of training [9] and these students need  to develop self-efficacy to use the skills learned when faced with a resuscitation scenario[2] . (paragraph removed)

As a consequence, the development of strategies that facilitate the learning of students within a university environment, can minimize the deleterious effects of the lack of knowledge on BLS, and these strategies can contribute to increase the chance of survival of the individual affected by CPA in public or private places. (paragraph removed)

The dissemination of knowledge and the implementation of measures that facilitate appropriate care routines for victims of CPA, must accompany the progress of Science, and the use of simulation in universities of Health Sciences is one of these measures, however this is something implated recently in Brazil and the investment for its realization involves infrastructure, such technology, trained teachers, which still makes the cost of implementation high [10].

According to a study that evaluated the effectiveness of direct instruction, the learning difficulty is not always limited to problems with the student, often the problem is in instruction [11](). The direct instruction model proves to be suitable for teaching BLS, where it is assumed that students can benefit from well-designed and unambiguous instructions. According to a study carried out in Australia, simulation training improves the confidence of the professionals involved, however, this does not mean improving the ability to perform CPR procedures [12].

The synthesis lecture on BLS (SLBLS), is part of the construction of the teaching-learning process, based on direct instruction, which aims to develop the cognitive part of it, characterized by the intellectual activity of the process, in order to promote initial learning about BLS. In this phase, students are concerned with how to start the procedures and with the actions related to the execution of the psychomotor skills that should be developed, showing that the acquisition, retention and transfer of knowledge, associated with the psychomotor skills, are essential to better care in the outcome of CPA [12-14]. (paragraph removed)

Once the low level of knowledge of Nursing, Physiotherapy and Medicine students in BLS is proven, which implies in not performing high quality CPR, resulting in a negative social impact and considering the new teaching proposals on the BLS, this study aims to evaluate the effect : To evaluate the performance of a synthesis lecture on BLS as a teaching strategy, among undergraduate students of medical sciences.

 [Methods]

  • Line 77: They do not need the abbreviation FMU.

Answer

Line 77 was modified, the abbreviation was replaced by  Faculdades Metropolitanas Unidas,

  • Line 85: “It would be better, “Medicine of 175, Nursing of 136 and Physiotherapy of 111”

Consisted of 422 university students from the first and fourth year of Health Sciences course, as follows, Medicine (175), Nursing (136) and Physiotherapy (111), with 281 being students of the first year and 141 from the fourth year of the respective courses.

Answer

Line 85 was modified

Consisted of 422 university students from the first and fourth year of Health Sciences course, as follows, 175 of  Medicine students, 136 of Nursing and 111 of Physiotherapy,  of which 281 of these people are taking first year and 141 doing the fourth year of their respective courses.

  • In Table 1: not “,”, but “.”

Answer: Table 1 was modified

Course

  Grade

Medicine

n (%)

Nursing

n (%)

Physiotherapy

n (%)

TOTAL

n (%)

First Year

132 (75.4)

89 (65.4)

60 (54)

281 (66.6)

Fourth year

43 (24.6)

47 (34.6)

51 (46)

141 (33.4)

TOTAL

175 (100.0)

136 (100.0)

111 (100.0)

400 (100.0)

  • Line 105: The word, AHA, has been shown in Introduction.

A questionnaire on BLS based on the guidelines published by the American Heart Association (AHA) was performed both in the pre-test and posttest stages to collect the data.

A questionnaire on BLS based on the guidelines published by the AHA was performed both in the pre-test and posttest stages to collect the data.

  • Line 124: What is country and city about Laerdal Medical?

The mannequins were from the Little Family Manikin Pack model and the AED  manufactured by Laerdal Medical.

Answer

The mannequins were from the Little Family Manikin Pack model (Laerdal Medical Corporation, New York, USA)and the AED (Laerdal Medical Corporation, New York, USA).

Line  125-146: They should describe not itemize style but usual style.

Answer: The process was described

The Lecture on Basic Life Support was administrated as follows: The SLBLS was carried out within a logical sequence, and initially the epidemiology of cardiac arrest in Brazil and worldwide was addressed, in addition to the survival rate after CPR, and then guidance on the survival chain was provided.

Next, instructions were given on how to recognize a victim in cardiopulmonary arrest, about the constituents of the BLS, demonstration of the locations for checking the central pulses according to age groups, a practical demonstration with the mannequin, of the correct opening of the airways in victims of trauma and non-traumatized, and how to provide adequate ventilation according to different age groups. Explanations about the locations and mode of cardiac compressions for babies were also given; children, adolescents with 1 and 2 rescuers present and with and without guaranteed advanced airway.

They were instructed on the characteristics of good quality CPR, where the use of the AED was reproduced and the peculiarities according to the age groups, the demonstration of the complete sequence of the CPR using the AED, the duration of the CPR cycles and on the reassessment of the CPR. patient.

Afterwards, the discussion of clinical cases was initiated, with the demonstration of the correct sequence of care in two different scenarios, one, after finding unconscious victims child and baby and another scenario where a sudden collapse was seen in an adolescent. The SLBLS ends with clarifying students' doubts before the post-test. Right after the class, the post-test was applied with the same conditions as the pre-test.

  • Line 163/169: They deleted the title of Variables and Sample loss.

Answer

The title of variables and sample loss are included in lines150-157

Variables

The variables analyzed were: the total number of correct answers in the pre and post-test, the difference in the number of correct answers between pre and post-test (dependent variables), the group of student's course, their grade course, and the presence of previous CPR training (independent variables).

Sample loss

There was a sample loss of 28 students (6.6%) due to not complying with the protocol as planned.

  • In Statistical analysis: As I mentioned, they do not need to make many new lines.

Answer:  No new lines were  added, but the paragraphs were rewritten

The data were descripted considering distributions by relative and absolute frequency, and measures of central tendency: average, median and fashion. The analytical treatment was performed by comparing means or medians using the Mann-Withney test (independent data), Willcoxon test and Kruskal-Wallis test (non-independent data). In addition, correlation analyzes were performed using Pearson's correlation coefficient.

 The level of significance adopted (α) 0.05. Interquartile range was used to compare the medians of pre and post-tests between first and fourth year students of Medicine, Nursing and Physiotherapy. The difference in correct answers between the pre and the post test had a normal distribution, with a median of 6.5 points. For data analysis, the following software was used: MedCalc version 17.2 and GraphPad version 6.01.

  • In Results: Also, they do not need to make many new lines.

Answer

No new lines were added, but lines 175/177 were rewritten:

Of the 422 students included in the final sample, 186 (44.1%) had previously undergone BLS training. In the knowledge assessment, the 422 students obtained a median of correct answers in the pre-test of 40.0%. In contrast, in the post-test, the median was 75.0%, p <0.0001, as is shown in figure 2.

No new lines were added, but lines 197/206 were rewritten:

An increasing trend is observed in the total number of correct answers between pre and post-test in all groups evaluated. This fact is corroborated by students' performance with the highest number of correct answers in the pre-test and evolved with an improvement in the post-test score. There was no statistically significant difference in comparing the medians of the number of correct answers in the respective courses (p = 0.1117).

On comparing the medians of correct answers in the post-test of the students of the respective courses, there was a significant statistical significance (p <0.0001) between medicine and nursing (p <0.001), medicine, and physiotherapy (p <0.01 ). But there was no statistical difference when comparing the Physiotherapy and Nursing courses (p> 0.05). According to their number of correct answers in the post-test, of the 422 students evaluated, 411 (97.5%)  improved performance.

  • Line 227 (table 2): I am unsure of the meaning of “a” in (7 a 7) and so on. Please rewrite to usual one.

Answer  

 Table 2 was modified as follows:

Grade

Median Pre-test

CI 95%

P-Value

Median post-test

CI 95%

P-Value

Median  difference of correct answers Post-Pre-test

P-Value

First-year

7 (35%)

(7-7) (35 -35%)

p<0.001

15 (75%)

(14-15) (70 -75%)

p<0.001

7 (35%)

p<0.001

Fourth-year

10 (50%)

(9 -11) (45-55%)

16 (80%)

(16 -17) (80-85%)

6 (30%)

  • Line 236 (table 3): Also, I am unsure of the meaning of “a” in “30 a 50” and so on. Please rewrite them.

Answer

Table 3 was corrected:

Course

Median correct answers pré-test (%)

interquartile range (p25 a p75)

P-Value difference between 1st and 4th year

Median  correct answers post-test (%)

interquartile range (p25 a p75)

P-Value difference between 1st and 4th year

Median  difference of correct answers  post test Vs. pre-test (%)

interquartile range (p25 a p75)

P-Value,  difference between 1st and 4th year

NURSING 1st year

40

30-50

p=0.0462

70

60-80

p=0.7042

(ns)

25

 15-40

p=0.1460

(ns)

NURSING 4th year

50

35-60

70

 60-80

25

10-35

PHYSIOTHERAPY 1st year

30

22.5-40

p=0.0001

70

60-75

p<0.001

32.5

 25-47.5

p=0.4450

(ns)

PHYSIOTHERAPY 4th year

45

40-55

80

80-85

35

 20-40

MEDICINE 1st year

35

25-45

p<0.001

75

70-85

p=0.0007

42.5

30-50

p<0.001

MEDICINE  4th year

55

41.3-65

85

 80-90

30

 20-38.8

[Discussion]

  • Line 237-345: Please decrease many paragraphs.

Answer

The Discussion was reduced, the texts in red were eliminated

 It was observed that the synthesis class model on the fundamentals of BLS, offered to the 422 students of health sciences was satisfactory. Indicating cognitive learning since most undergraduate students in Nursing, Physiotherapy, and Medicine practically doubled its indicators compared pre- and post-test evaluations. The median of their post-test results approached the performance result recommended by the American Heart Association (AHA) standard course on BLS (84%).

 In Pakistan, a study showed that knowledge of first aid is something that medical students have not given due importance with consequent effect on the quality of performing CPR for newly qualified doctors; despite the importance of training, BLS now has worldwide recognition (15).

 The prior knowledge of health sciences students about BLS was 40%, which is similar to the findings of a study on university students' knowledge about BLS. Correct answers were frequent for 35.8% of questions about BLS (8).

 In France, a study carried out with medical students from more advanced years showed that they do not feel prepared to conduct BLS and that only 1/3 of the students reported being able to provide this assistance. Using an instrument with objective self-perception questions about BLS, more than 2/3 of the students reported not being qualified or having insufficient qualification, while only 2% reported feeling qualified (7).

 In this research, trained students and those without previous BLS training obtained low pre-test scores, below the 84% recommended AHA score. These findings point to a cognitive deficit on the necessary procedures for adequate high-quality BLS, denoting the need for permanent and continuing education on this significant theme.

 Studies comparing the theoretical performance between students with and without prior training showed better theoretical results than previous training. These results converge to those of the present research, indicating that previous training indicates students' better BLS (15-18).

 In this study, students from the first year of medicine showed a significant difference from the initial and final evaluations. They were the group that obtained the highest benefit of the SLBLS. The first-year nursing students were the group with the most significant number of individuals who reported having previous knowledge about BLS. These students remained with satisfactory performance in the post-test, but with a smaller difference in correct answers between the pre and post-test; that is, it was the group who benefited less from the class.

 In India, a study evaluated the retention of knowledge and skills transmitted to first-year medical students about BLS. The pre-test showed unsatisfactory results and improvement in the post-test after demonstration training. It corroborates with the results of this current study, which showed improved cognitive knowledge in the three courses after the BLS lecture (19). Other studies carried out with first-year students showed results similar to those of the present study (15, 17, 20).

 The analysis of students' pre-test performance from the fourth year of the Nursing, Physiotherapy, and Medicine courses showed that the theoretical knowledge on BLS was low, indicating insufficient knowledge, even in these university students of a more advanced course level.

 Overall, students from the fourth year of the courses performed better in the post-test, attributed to previous training outside the university environment or BLS activities at the university. Correctly, as for the nursing course students, who obtained excellent performance both in the pre-test and in the post-test, these results can be attributed to the conditions said before and the fact that most of them were technical professionals in the field of Nursing.

 Our study reveals that the first-year students had the highest gain concerning previous knowledge. However, they were not the ones who had the highest score in the final evaluation, which is attributed to more advanced students with previous training, emphasizing the retention of knowledge in the face of exposure.

 A study reported that young doctors do not perform adequate CPR due to university teaching failure.  Their opportunities for training in the workplace were remote, showing that BLS education strategies should be reviewed at universities (21). In another study in Romania, they share the idea of including BLS training in the university curriculum (22).

 A fifty-year review of the effectiveness of direct instruction stated that students interpret the information they are given. However, their learning only occurs when the information presented is explicit, logically organized, and sequenced (11).

 When students are actively involved in learning, they are more influenced by domains, the cognitive domain related to the content domain, and the psychomotor domain related to practical skills (23-25). Consequently, continuing education in BLS can reinforce this learning, providing greater confidence to students, which would increase their chance of performing high-quality CPR in the face of a real event.

 According to this study's protocol, the demonstration of a BLS lecture aims to provide prior instruction on the execution of motor skills, enabling the student to obtain information about the nature of the task to be performed, evidencing the "how to do" process. The purpose of these lectures is to stimulate students at the beginning of the course in the cognitive part and as for students of a more advanced level to contribute to the reinforcement of knowledge since cognitive learning is a premise for the future development of psychomotor skills.

 According to  AHA guidelines on medical education, the literature on resuscitation education has been limited to results that focus on a short learning period(2). With the model of synthesis BLS lecture presented in the current study, it was possible to achieve positive results, compared to those in the literature, proving to be very relevant, especially for students at the beginning of the course.

 A study carried out in the United Kingdom reported that students improved the post-test; it is similar to the results of the present study. However, on analyzing the students who worsened or continued with the same previous knowledge, there was a smaller number (2,7%)  in our study than the UK research, where 29% worsened (18).

 University students in the health field do not have enough knowledge about BLS  to perform high-quality CPR. It is a serious reflection of the failure to teach this topic in universities. These institutions need to invest in better realistic training centers and continuing education, making it possible to improve knowledge retention and transmit confidence to the student, who currently takes a deficit of knowledge from the university to practical life.

 The success of resuscitation focuses on the correct performance of specific steps. In the case of students, who have not been frequently exposed to situations that indicate it, the synthesis class, with demonstration based on scenarios, favors cognitive knowledge acquisition. It excites the first-year student by demonstrating a CPR simulation, studying the theory on the subject, and preparing for future BLS skills training.  (Paragraph removed)

 In the sample of this study, the performance of students of health sciences was sufficient to show learning in the cognitive part, from the BLS lecture's execution. Most students showed improvement in their indicators compared between the pre and post-test assessments. The students who benefited the most from the training were first-year medical students. However, the highest grades were obtained by the fourth-year medical students. The undergraduate course that benefited the least was the Nursing course, where the first students year have already demonstrated more excellent prior knowledge about BLS.

 The traditional teaching method of BLS based on translational Science is the best to convey confidence to the student. The complete course is based on realistic simulation, allowing the student to acquire learning cognitive and develop practical skills. However, at different times of their course, to improve cognitive learning functions, with more frequent training, having favored the short time of accomplishment,  the synthesis lectures on BLS find its space in education.

 To the best of the author's knowledge, no studies compare the performance of this type of educational activity in undergraduate students of health sciences in three different institutions in Brazil's two distant regions. The north region and southeast region have significant differences in human development. Furthermore, this initiative is economically viable with an excellent cost-benefit ratio for the universities, considering that the traditional method training on BLS following AHA guidelines requires an average of eight hours.

 Our findings showed a significant improvement in cognitive learning and provided initial learning about BLS for undergraduate students.
